# Impaired gastric and urinary but preserved cardiac interoception in women with endometriosis

**Chiara Cantoni** [1*], **Sofia Ciccarone**[1], **Maria Grazia Porpora**[2], **Salvatore Maria Aglioti**[1,3,4]

**1** Department of Psychology, Sapienza University of Rome, Rome, Italy, **2** Department of Maternal, Infantile and Urological Sciences, Sapienza University of Rome, Rome, Italy, **3** Department of Psychology, Sapienza University of Rome & Center for Life Nano- & Neuroscience, Italian Institute of Technology (IIT), Rome, Italy, **4** IRCCS Santa Lucia Foundation, Rome, Italy

* chiara.cantoni@uniroma1.it

## Abstract

Endometriosis is a chronic inflammatory gynaecological condition frequently associated with chronic pelvic pain. Visceral hypersensitivity could be present, like in other chronic pain conditions, causing altered levels of interoception. So far, studies have explored interoceptive deficits in chronic pain individuals mainly using questionnaires or cardiac interoceptive accuracy tasks. Here, we explore the cardiac, gastric, and urinary domains to probe interoceptive differences between patients with endometriosis and healthy women. 30 patients and 30 controls underwent three interoceptive tasks for assessing the cardiac domain (using the Heartbeat Counting Task, HCT), the gastric domain (using the Water Load Test-II, WLT-II) and the bladder domain (using a novel Urinary Interoceptive Task, UIT). Participants also completed bladder interoceptive beliefs measures and subjective pain ratings for each endometriosis symptom (dysmenorrhea, dyspareunia, dyschezia, chronic pain). A positive correlation between the WLT-II and the UIT emerged in all participants (R = 0.47, p < 0.001), indicating that the lower the gastric interoceptive abilities, the lower the urinary ones. Moreover, compared to healthy controls, women with endometriosis exhibited lower scores in the WLT-II (t(58) = 4.6814, p < 0.001) and the UIT (t(39.931)= 5.1462, p < 0.001), as well as higher scores in the subjective bladder beliefs questions (t(57.346)= -4.0304, p < 0.001). Results indicate a dissociation between patients' poor objective performance on interoceptive tasks and their high bladder interoceptive beliefs. UIT scores were associated with pain symptoms, suggesting that patients, probably due to sensitisation, struggle to ignore discomfort sensations, resulting in reduced accuracy in detecting physiological signals coming from the pelvic area.

**Data availability statement:** The datasets generated and analysed during the current study and supporting information, and the components of the research methodology needed to reproduce the reported procedure(s) and analyses are publicly available on the Open Science Framework repository (DOI: 10.17605/OSF.IO/H7V5Y) and on Figshare here: https://figshare.com/projects/INTENDO_INTeroception_and_ENDOmetriosis_/236981

**Funding:** This work was supported by Ministero dell'Università e della Ricerca (PRIN n. 20229JPNT7, https://prin.mur.gov.it/) (to S.M.A.). The funder had no role in study design, data collection and analysis, decision to publish, or preparation of the manuscript.

**Competing interests:** The authors have declared that no competing interests exist.

## Introduction

Endometriosis is a chronic oestrogen-dependent disease characterised by the presence of endometrial tissue, that physiologically lines the uterine cavity, in ectopic sites [1], causing chronic inflammation [2]. This condition affects 10%-15% of women of reproductive age [3], causing different instances of pain, specifically dysmenorrhea (menstrual pain), dyspareunia (pain during sexual intercourse), dyschezia (pain during defecation) and chronic pelvic pain [4]. Medical treatment includes analgesics and long-term use of hormonal drugs [5] but, in the most severe cases, surgical treatment may be necessary [6,7]. Unfortunately, the disease is diagnosed with an average delay of 7 years from the onset of symptoms [8–11]. It is often mistaken for conditions with similar symptoms such as pelvic inflammatory disease, irritable bowel syndrome, interstitial cystitis, and fibromyalgia [12].

Generally, individuals suffering from chronic pain and visceral hypersensitivity (i.e., increased sensitivity to pain originating from the internal organs) typically exhibit altered interoception, i.e., they perceive the physiological state of their body differently if compared to healthy controls (i.e., lower interoceptive accuracy [13–21] or higher interoceptive accuracy [22]). These individuals may show deficits in one of the dimensions of interoception defined by the multi-dimensional interoceptive framework by Suksasilp & Garfinkel (2022) [23]: *interoceptive accuracy*, i.e., the correspondence between objectively measured physiological events and individuals' reported experience of those events, ascertained through behavioural tests; *interoceptive beliefs*, i.e., measures of beliefs, both available to and beyond conscious access, concerning individuals' interoceptive sensations and experiences. Includes self-report measures, such as questionnaires and confidence ratings, and task-based measures of (implicit) prior beliefs thought to influence interoceptive perception; and *interoceptive insight*, i.e., meta-cognitive evaluation of experience/performance, e.g., the correspondence between accuracy during an interoceptive task, and (self-reported) perceived accuracy or confidence during the task (for an exhaustive description of all the different dimension of interoception please refer to[23]).

For instance, patients with chronic pain conditions such as fibromyalgia [24,25] and irritable bowel syndrome [26,27] showed altered cardiac interoceptive accuracy - measured using the Heartbeat Counting Task [28] - and beliefs - assessed via self-report questionnaires - if compared to healthy individuals [29]. Thus, endometriosis, characterised by chronic pain and visceral hypersensitivity [30–32], may also be associated with interoceptive alterations. Recent studies using subjective assessments have demonstrated that interoceptive self-regulation mediates the relationship between pain and depression in endometriosis patients, underscoring the crucial role of interoception in this condition [33]. Nevertheless, Desmedt and colleagues (2022) [34] suggest that uncertainty remains about whether current "interoceptive sensibility" and "self-report interoceptive scales" measures really assess a common construct.

Building on this framework, this study employs a series of interoceptive accuracy tasks to comprehensively map the interoceptive abilities of endometriosis patients across three distinct body domains.

Overall, we aimed to investigate interoceptive alterations in the three body domains and their association with symptoms severity. Patients and healthy controls performed the Heartbeat Counting Task to assess the cardiac domain [28], the Two-step Water Load Test for the gastric domain [35] and a novel version of a urinary interoceptive task designed to test bladder interoception. However, it is important to acknowledge the limitations of the Heartbeat Counting Task (HCT, [28]), which has been widely used to measure interoceptive cardiac accuracy. Recent evidence has raised concerns about the validity of the HCT, suggesting that it may not purely assess interoceptive accuracy but could also be influenced by other factors such as general cognitive abilities, beliefs about heart rate, and knowledge of cardiac rhythms. For instance, Desmedt et al. (2023) highlighted that the HCT results may reflect participants' expectations or their ability to estimate time rather than their capacity to accurately perceive cardiac signals [36]. These limitations underscore the importance of complementing this cardiac task with additional interoceptive tasks that assess other domains and provide a more comprehensive assessment of interoceptive abilities. We focused on the bladder due to its location in the pelvic area, patients' focal pain area. To the best of our knowledge, the already existing bladder sensitivity tests are either invasive [37], primarily focused on pain rather than bladder interoceptive signals [38] or require gynaecological equipment, such as sonography, to be performed [39,40]. Thus, we conceived a version of the urinary task - inspired by an already existing water load diuresis protocol [39,41,42] - to measure bladder interoception specifically in an accessible and non-invasive way. Detecting interoception alterations in endometriosis could help identify possible markers of the disease and provide potential non-invasive diagnostic tools. Moreover, since endometriosis symptoms may affect the way women see themselves, we also investigated the relationship between symptoms of endometriosis and women's body image [43,44].

## Materials and methods

### Participants

We recruited thirty women diagnosed with endometriosis (mean = 32.93 ± 6.58 yr., range = 22–44 yr.) and thirty healthy controls individually matched for sex and age (mean = 31.6 ± 6.88 yr., range = 21–45 yr.). The experimental protocol was reviewed and approved by the ethics committee of Sapienza University of Rome (Prot. n. 0000600, 28/03/2022) and followed the ethical standards of the 2013 Declaration of Helsinki. All the participants provided their written informed consent, they were naïve to the research purpose and received 20€ as reimbursement for their participation. The recruitment period took place between 28.03.2022 and 05.05.2022. Demographic data are presented in Table 1.

**Patients.** Patients either referred to the Endometriosis and Pelvic Pain Outpatient Clinic of the Maternal, Child, and Urological Sciences Department of the Policlinico Umberto I of Rome or were members of the *Alice O.D.V.* Italian Association for Endometriosis. All patients had been diagnosed by specialised hospital services and were individually screened by the gynaecologist (M.G.P.) according to the following criteria.

**Table 1. Descriptive statistics.**

| Demographics and Endometriosis Variables | | | | | | | |
|---|---|---|---|---|---|---|---|
| | Hc (N = 30) | | Pz (N = 30) | | | | |
| | M | SD | M | SD | t | df | p |
| Age | 31.60 | 6.88 | 32.93 | 6.58 | -0.767 | 58 | 0.446 |
| BMI | 22.83 | 3.12 | 21.66 | 2.72 | 1.55 | 58 | 0.126 |
| VAS dysmenorrhea | 41.83 | 27.60 | 63.37 | 37.67 | -2.53 | 58 | 0.014 |
| VAS dyspareunia | 9.67 | 19.12 | 49.87 | 32.36 | -5.86 | 58 | 0.001 |
| VAS dyschezia | 15.90 | 25.26 | 50.97 | 36.12 | -4.36 | 58 | 0.001 |
| VAS chronic pelvic pain | 2.03 | 6.42 | 52.67 | 34.59 | -7.88 | 58 | 0.001 |

Means = M, standard deviations = SD of demographic and Endometriosis Pain variables for the healthy controls (Hc) and patients with Endometriosis (Pz) groups, and results of the t-tests. N.B.: BMI = Body Mass Index; VAS = Visual Analogue Scale.

*Inclusion criteria*: diagnosis of endometriosis according to the guidelines of the Italian Society of Gynaecology and Obstetrics [45] and presence of dysmenorrhoea, dyspareunia, chronic pelvic pain, and dyschezia.

*Exclusion criteria*: pacemaker; chronic intake of drugs (except for those taken to treat endometriosis); pregnancy; diagnosis of cardiac or cardiovascular diseases, neoplasia, neurological or psychiatric disorders, urinary disorders or lower urinary tract symptoms; presence of endometriosis adhesions on the bladder. Patients' clinical information is presented in Table 2.

**Healthy controls.** Healthy participants of the same age as the patients in the experimental group were recruited via advertisements on social networks. Exclusion criteria for the control group were: diagnosis of gynaecological, urological and gastrointestinal disorders, cardiac or cardiovascular pathologies, neurological or psychiatric disorders; current pregnancy; pacemakers; drug intake.

An a priori Power Analysis, based on previous literature [46,47], was carried out to determine the sample size using the G*Power 3.1.9.4 software [48], setting the following parameters: test family = F tests; statistical test = ANOVA fixed effect, one-way; effect size f = 0.4 ($\eta^2$ = 0.14); Power = 0.80; α err prob. = .05; number of groups = 2; Critical F = 4.03. The Power analysis suggested recruiting 26 individuals per group.

## Procedure

Upon arrival, participants provided their informed consent and started the experimental session with the gastric interoceptive task [35], which lasted approximately fifteen minutes. Subsequently, they were administered the Heartbeat Counting task [28] which lasted approximately five minutes. Afterwards, to ensure an adequate interval between the gastric and urinary water load procedures participants performed a virtual reality task, unrelated to the present project, which lasted about 30 minutes.

Participants were then asked to empty their bladder in preparation for the urinary water load. Notably, the task administration did not start until the bladder was emptied. Participants completed the self-report questionnaires throughout the urinary task, particularly during the period leading up to the maximum urinary stimulus (see paragraph below). Additionally, participants diagnosed with endometriosis were asked about their medical history, including the use of hormonal production-blocking medications or prior surgical interventions related to endometriosis. After completing the urinary interoception task, participants were debriefed about the purpose of the experiment. The entire experimental session (Fig 1) lasted approximately three hours.

## Self-report questionnaires

We administered the Italian versions of the following self-report questionnaires: the *Multidimensional Assessment of Interoceptive Awareness II* (MAIA-II [49]); and the *Body Uneasiness Test* (BUT [50]).

**Table 2. Patients' clinical information.**

|  | Patients (N = 30) | | |
|---|---|---|---|
|  | M | SD | % |
| Age | 32.9 | 6.58 | / |
| BMI | 21.66 | 2.72 | / |
| Endometriosis Surgery (%) | / | / | 53.33% |
| Progistinic (%) | / | / | 40% |
| Oestrogen (%) | / | / | 26,67% |
| Previous Pregnancy (%) | / | / | 26,67% |

M = mean; SD = standard deviation; % = percentage on the total of patients.

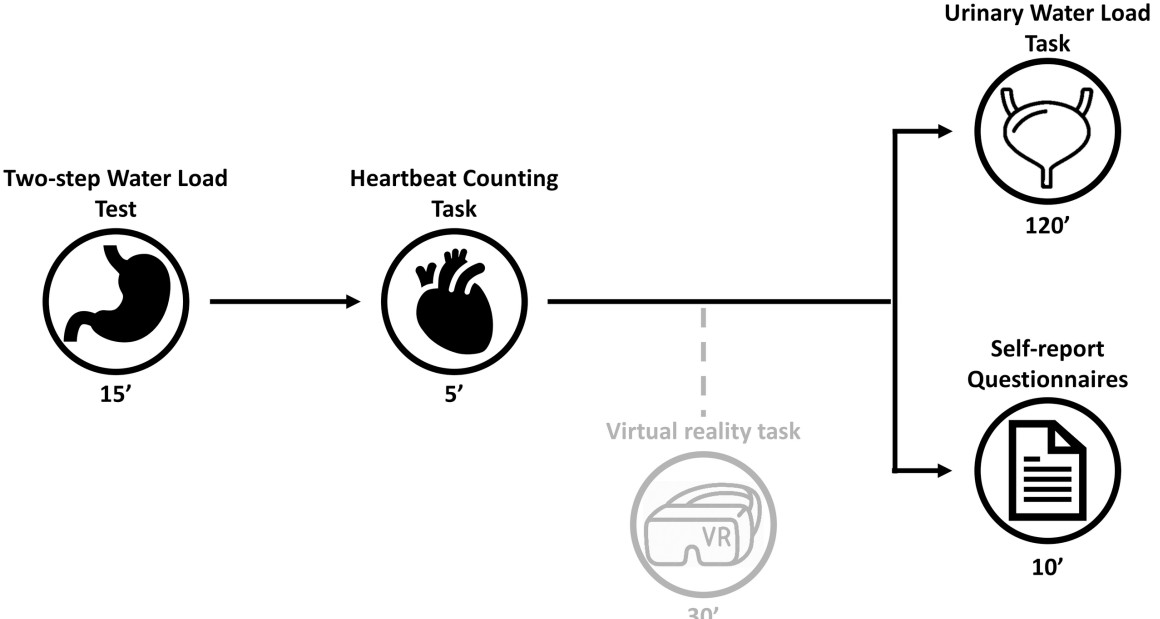

**Fig 1. Graphical description of the experimental timeline.** Note: the 'incidental' virtual reality task was not part of this project and in the context of the present research merely ensured that sufficient time elapsed between gastric and urinary water loads.

Finally, for each symptom of endometriosis (i.e., dysmenorrhoea, dyspareunia, dyschezia, and chronic pelvic pain), the patient's subjective perception of pain was measured using a visual analogue scale (VAS), where 0 corresponded to "no pain" and 100 corresponded to "worst pain imaginable" [51]. See further information in the Supplementary materials.

### Interoceptive tasks

The following tasks were used to assess interoceptive accuracy (i.e., the correspondence between objectively measured physiological events and individuals' reported experience of those events):

*Gastric Task* - Two-step Water Load Test (WLT-II [35]): Participants arrived at the laboratory having fasted from food for 3 hours and from water for 2 hours. The task involved two phases (taken from van Dyck and colleagues [35]): in the first phase, lasting a maximum of five minutes, participants were instructed to drink water until they felt the sensation of satiety (sat_ml); in the second phase, immediately after the first one and lasting a maximum of five minutes, participants were asked to drink water until they reached the point of maximum gastric fullness (Δfull_ml). At the end of the two phases, the ratio between the volumes of water ingested in the two phases is calculated to obtain an individual gastric interoception index (sat_%=(sat_ml/total_ml)*100). The index obtained is a number between 0 and 100, dimensionless and therefore independent of the participant's stomach volume: the higher the index, the higher the subject's gastric interoceptive accuracy. At the beginning of the task and after each drinking phase, participants were asked questions about their gastric sensations (See Tables S1, S2 in S1 File). This task lasted approximately 15 minutes (Fig S2 in S1 File).

*Cardiac Task* - Heartbeat Counting Task (HCT [28]): participants were asked to count their heartbeats during four-time intervals of different durations (25s, 35s, 45s, 100s), presented in randomised order, and to report the number of heartbeats they were able to perceive. During the task, the electrocardiogram (ECG) was recorded via three pre-gelled, disposable Ag/AgCl 50 mm electrodes arranged in a bipolar lead II configuration. Cardiac interoceptive accuracy was computed for each participant with the following formula: $1/4\Sigma(1-(|actual\_heartbeats-reported\_heartbeats|)/actual\_heartbeats)$, with higher scores indicating higher accuracy. The task lasted approximately 5 minutes (Fig S1 in S1 File).

*Novel Urinary Interoceptive Task* – (Urinary Interoceptive Task - UIT) (Fig 2). Participants started the task with an empty bladder (they were asked to urinate beforehand). They were then asked to drink 250 millilitres of water every 15 minutes for an hour, for a total of one litre in one hour. Then, they were asked to wait until they reached the *maximum urinary stimulus [void1_{ml}]*, described as *"the sensation you feel when it's impossible for you to hold the urine any longer"*. When this condition occurred, we asked them to urinate in a sterile container and we measured the millilitres of expelled urine on a digital scale. After this first step, we assumed that the participants' bladder was empty again. We then asked them to drink 250 ml of water within 15 minutes (to reactivate the urinary stimulus) and indicate the *minimum urinary stimulus [void2_{ml}]*, described as *"the first desire to void when you perceive the first stimulus to urinate, and you say, 'I could go but it's not necessary'"*. When participants reached the minimum urinary stimulus, we asked them to void the bladder and measured the millilitres of expelled urine. At this point, we assumed that the participant's bladder was empty once again. For a consistency check, we repeated this second step of the task once again, i.e., we asked them to drink 250 ml of water within 15' and to indicate the minimum urinary stimulus *[void3_{ml}]*. When participants reached the minimum stimulus again, we asked them to void the bladder one last time and measured the millilitres of expelled urine. Each time, before drinking and upon reaching the maximum and minimum stimuli, participants were asked to rate their bladder sensations with a VAS from 0 (indicating no bladder stimulus at all) to 10 (indicating the maximum urinary stimulus). Although a specific VAS score was not predefined as a minimum stimulus (i.e., each participant could indicate the desired number as minimum stimulus, following their own sensations), participants were instructed to consistently apply the same subjective criterion in both the void2ml and void3ml conditions (i.e., for each participant the VAS score assigned to void2ml as minimum stimulus should have been the same in void3ml). Finally, participants were asked five questions assessing their attitude towards bladder stimuli in their everyday life (See Tables S3–S5 in S1 File).

We created an index to assess bladder interoception, firstly by calculating the difference between the millilitres expelled during the first minimum stimulus and the millilitres expelled during the second minimum stimulus. This was necessary to assess a person's consistency in indicating her minimum stimulus. The *Urinary Interoceptive Index* was derived by the

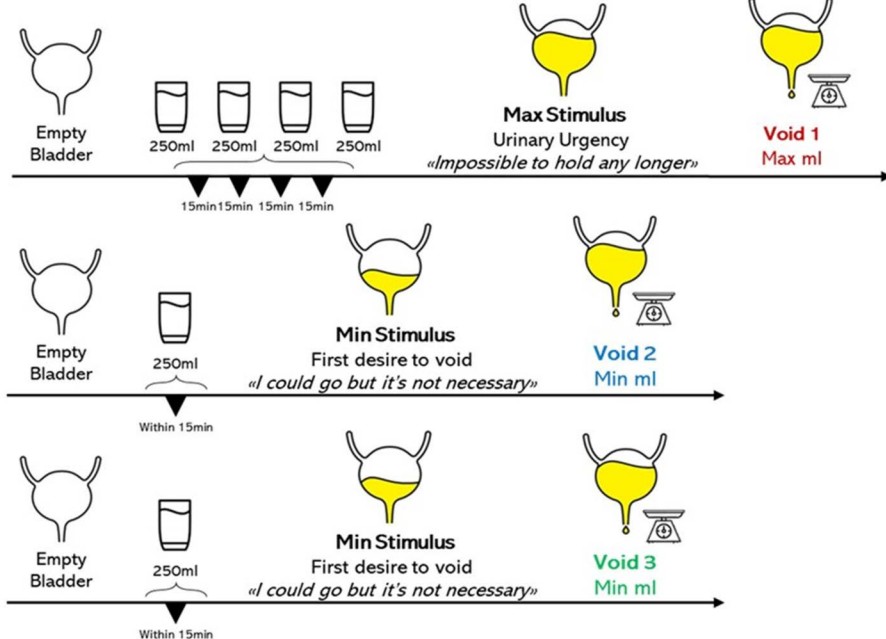

**Fig 2. Novel Urinary Interoceptive Task.** Graphical description of the novel Urinary Interoceptive Task.

ratio between the mean of the millilitres expelled at the two minimum stimulus voids and the millilitres expelled at the maximum stimulus void, subtracted from 1.

$$\left[1 - \frac{(void2ml - void3ml)/2}{void1ml}\right]$$

The bigger this ratio, the better the subjects are at discriminating between two different stimuli coming from their bladder. The task duration was approximately two hours, contingent on the participants' bladder activity.

### Data analysis

**Relationship between the interoceptive tasks.** Since the Urinary Interoceptive Task was a new protocol, we wanted to explore if it correlated with existing validated interoceptive tasks taking into account the whole sample (N = 60). Thus, we ran three correlation analyses using a correlation package (v0.6.0; [52]) to test the relation between the Urinary Interoceptive Task, the Heartbeat Counting task, and the Two-step Water Load Test. Correlations were calculated using Pearson's r. We also wanted to explore if participants' consistency in discriminating the two minimum urinary stimuli correlated with their ability to detect gastric and cardiac stimuli, measured through the gastric Waterload test and the Heartbeat Counting Task, respectively. Thus, we ran two correlation analyses using a correlation package (v0.6.0; [52]) to test the relation between participants' consistency, the Heartbeat Counting task, and the Two-step Water Load Test.

**Interoceptive accuracy and beliefs in patients.** To test if cardiac, gastric, and urinary interoceptive scores were different between patients with endometriosis and healthy controls we ran three t-tests: for the urinary interoceptive task, we ran a Welch Two sample t-test since the data did not meet the assumptions of homogeneity of variance, while for cardiac and gastric interoceptive tasks we run two separate Student's t-tests. To test the group differences in interoceptive beliefs we run a series of t-tests for each subscale of the Multidimensional Assessment of Interoceptive Awareness II (i.e., parametric t-tests for Body Listening, Not Worrying, Not Distracting, Self-Regulation and Attention regulation subscales as they met all the assumptions). As for the five questions assessing participants' bladder beliefs, we first ran a series of Wilcoxon rank-sum tests on each question assessing attitude towards bladder stimuli between the control and patient groups, controlling for multiple comparisons with False Discovery Rate (FDR) correction [53]. To test participants' consistency in detecting the two minimum stimuli (i.e., void2 and void3) in the two groups, we performed two non-parametric t-tests.

**Pain and urinary interoception.** We explored the differences between patients with endometriosis and healthy controls in pain perception by running a Mann-Whitney non-parametric test for each pain VAS (i.e., dysmenorrhoea, dyspareunia, dyschezia, chronic pelvic pain). We then ran a moderation analysis using Jamovi software.

**Body image questionnaire.** We explored the differences between patients with endometriosis and healthy controls in all the different subscales of the questionnaire (i.e., Body Uneasiness Test) by running a series of t-tests.

## Results

### Relationship between the interoceptive tasks

The correlation analyses performed between the urinary, cardiac and gastric interoceptive tasks showed that, considering the whole sample, the novel Urinary Interoceptive Task positively correlated (R = 0.47, p = 0.00015; Fig 3) with the Two-step Water Load Test [35], but both tasks did not correlate with the Heartbeat Counting Task [28] (R = 0.25, p = 0.055; R = 0.17, p = 0.21).

The correlation analyses performed considering the whole sample between participants' consistency in detecting their minimum urinary stimuli (measured as the difference between void2 and void3) and cardiac and gastric interoceptive

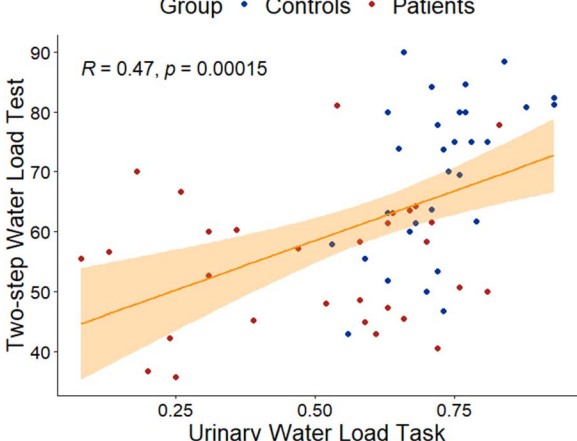

**Fig 3. Correlation between the Two-step Water Load and the Urinary Interoceptive Task.** The higher the participants' gastric interoceptive accuracy (i.e., higher scores indicate higher ability to perceive gastric signals), the higher their urinary interoceptive accuracy (i.e., higher scores indicate higher ability to perceive urinary stimuli). Running correlation analyses separately for each group (healthy controls vs. patients with endometriosis) revealed that the novel urinary task did not correlate with the Heartbeat Counting Task (R = 0.2389, p = 0.2034) but did correlate with the Water Load Test (R = 0.5522, p = 0.001557) in healthy controls. In contrast, in the endometriosis group, there were no significant correlations with either the Heartbeat Counting Task (R = 0.1516, p = 0.4239) or the Water Load Test (R = 0.1357, p = 0.4745).

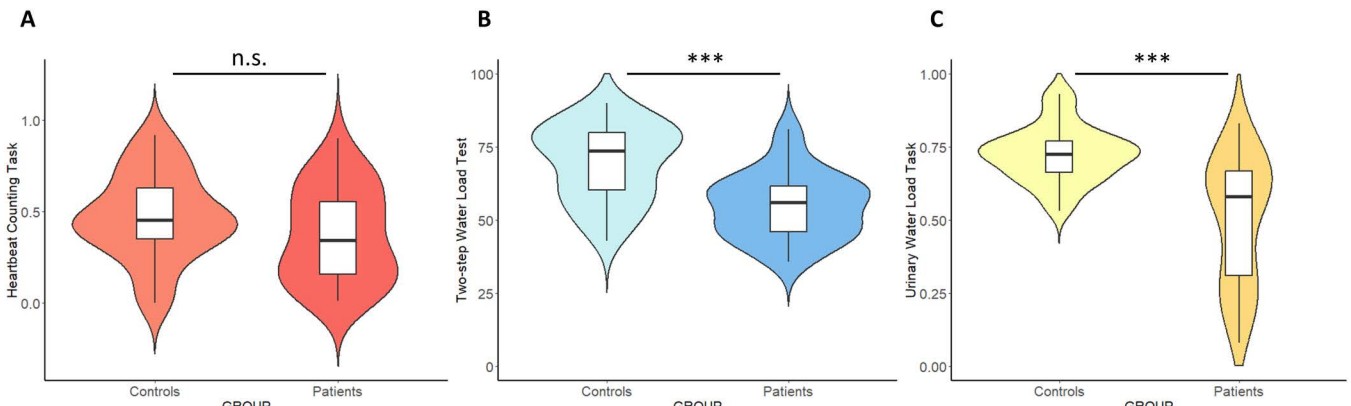

**Fig 4. Interoceptive Accuracy Tasks.** T-test to explore differences in the a) cardiac interoceptive task between patients with endometriosis and healthy controls; b) gastric interoceptive task between patients with endometriosis and healthy controls; c) urinary interoceptive task. Higher scores indicate a greater ability to detect interoceptive signals (i.e., cardiac, gastric and urinary). Significance: '***' p ≤ .001; 'n.s.' non-significant results.

tasks showed that participants' consistency in detecting their minimum urinary stimuli did not correlate with participants' ability to detect either their gastric (R = -0.096, p = 0.46) or cardiac (R = -0.03, p = 0.82) signals.

Further analyses of correlations between the tasks in the two groups separately are presented in the Supplementary Materials.

## Interoceptive accuracy and beliefs in patients

Two independent t-tests were carried out to explore the differences in cardiac and gastric interoceptive accuracy between women with endometriosis and healthy controls. No difference emerged between the two groups in the Heartbeat

Counting Task (t(58) = 1.5654, p = 0.12291) ([Fig 4a]). In contrast, a significant difference emerged in the Two-step Water Load Test (t(58) = 4.6814, p < 0.001) ([Fig 4b]), indicating that healthy participants are more accurate than patients with endometriosis in the gastric, but not in the cardiac interoceptive accuracy task. The Welch t-test to examine the difference between patients and healthy controls in the Urinary Water Load test was significant (t(39.931)= 5.1462, p < 0.001) ([Fig 4c]), suggesting that healthy controls are more accurate than patients in the bladder interoceptive task.

To make sure that patients' low gastric and bladder interoceptive accuracy were not due to differences related to different treatments of endometriosis, we checked for any differences between patients who had or had not undergone surgery; no differences in gastric (t(28)= -0.24944, p = 0.8048) or bladder (t(28)= -0.66266, p = 0.513) interoceptive accuracy emerged in these two subgroups. We also controlled for any hormonal medication taken (i.e., progestin, oestrogen, or no medication), and found that no significant differences in gastric (F(2,27)=1.516; p = 0.238) or bladder (F(2,27)=1.398; p = 0.265) interoceptive accuracy emerged in these three subgroups. We also checked that any differences between patients with endometriosis and healthy controls in bladder interoceptive accuracy were not due to differences in urinary incontinence or urinary retention. Therefore, we measured whether there was a difference between patients and control subjects in the time required to reach the maximum stimulus (VAS = 10). No significant differences (t(58)=1.7138, p = 0.09191) emerged.

We also performed two non-parametric t-tests to test whether void2 and void3 were consistently different in the two groups. Results showed that void2 did not significantly differ from void3 neither in the control group (W = 300, p = 0.171) nor in the patients' group (W = 265, p = 0.510).

To assess differences in bladder interoceptive beliefs, we performed a series of Wilcoxon rank-sum tests on each question assessing attitude towards bladder stimuli between the control and patient groups (see [Fig 5]). To control for multiple comparisons, we performed the False Discovery Rate (FDR) correction. The adjusted p-values are as follows: Urinary attention: 0.0449; Urinary interference: 0.0212; Urinary worry: 0.0071; Urinary night-time frequency: 0.0449; Urinary day-time frequency: 0.0317. A significant difference was found for urinary attention (W = 317, p = 0.0449, FDR-corrected), urinary interference (W = 273, p = 0.0085, FDR-corrected), urinary worry (W = 237.5, p = 0.0014, FDR-corrected), urinary nighttime (W = 318.5, p = 0.0427, FDR-corrected), and urinary daytime (W = 293, p = 0.0190, FDR-corrected).

To test differences in interoceptive beliefs between the two samples, we conducted a series of t-tests, one for each MAIA-II subscale: none of these revealed a statistically significant difference between the two groups (all p's > 0.05; See Supplementary Materials), indicating no across group differences in interoceptive beliefs.

## Pain and urinary interoception

We performed four non-parametric Mann-Whitney tests to probe any differences between the groups in subjective pain perception. As expected, patients with endometriosis reported experiencing significantly more pain than healthy women in all four pain subscales: dysmenorrhoea (U = 261.5, p = 0.005), dyspareunia (U = 138.5, p < 0.001), dyschezia (U = 218, p < 0.001) and chronic pelvic pain (U = 57.5, p < 0.001).

To further explore the relationship between chronic pelvic pain and urinary interoception, we ran a moderation analysis with "Group" (i.e., patients with endometriosis vs. healthy controls) as a moderator, urinary interoception as a dependent variable, and the chronic pelvic pain perception (assessed via pain VAS scores) as a predictor. This analysis revealed a marginally significant effect of the moderator (group) on the relationship between urinary interoception and chronic pelvic pain (Estimate: -0.00440 (SE = 0.00228, Z = -1.93, p = 0.054).

## Body image questionnaire

Results of the t-tests for each subscale of the Body Uneasiness Test revealed a statistically significant difference in the Depersonalization subscale (t(46.951)=-2.2049, p = 0.0324), suggesting that patients with endometriosis have a higher tendency to experience detachment and estrangement toward their own body if compared to healthy controls. None of

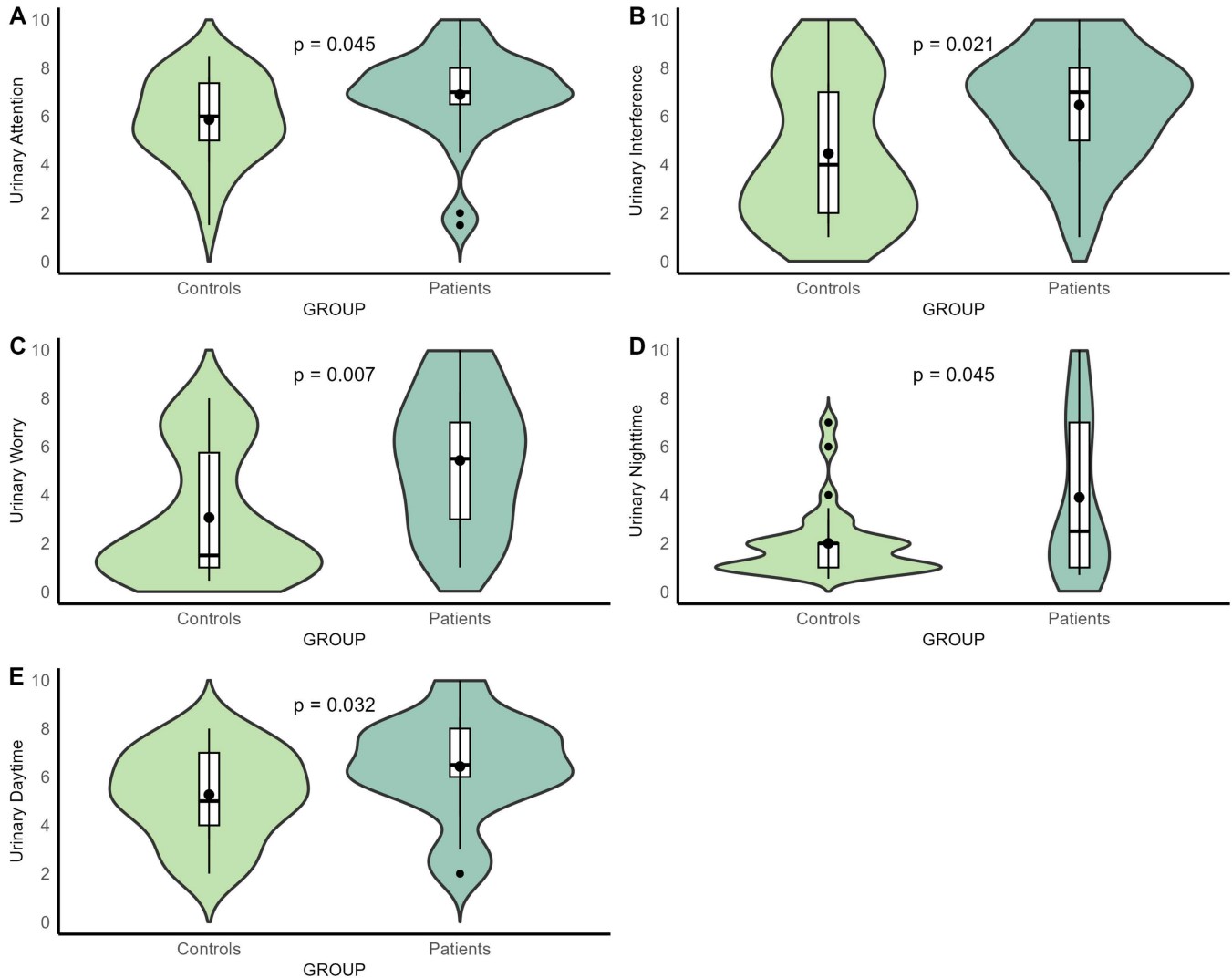

**Fig 5. Bladder Interoceptive Beliefs.** Non- parametric t-tests to explore differences in a) attention to bladder stimuli between patients with endometriosis and healthy controls (higher scores indicate greater attention towards bladder stimuli); b) interference of bladder stimuli with daily activities (higher scores indicate greater interference of bladder stimuli); c) concern towards bladder stimuli between patients with endometriosis and healthy controls (higher scores indicate greater concern towards bladder stimuli); d) nighttime need to empty the bladder (higher scores indicate higher need to empty the bladder during nighttime); e) daytime need to empty the bladder (higher scores indicate higher need to empty the bladder during daytime).

the other analyses performed on the other subscales resulted statistically significant (all p's > 0.05; See Supplementary Materials).

## Discussion

Studies investigating interoceptive abilities in chronic pain conditions have mainly employed interoceptive belief questionnaires or cardiac interoceptive tasks [17,24,54]. Bladder sensations have been measured in chronic pain conditions (such as IBS [38]) and primary dysmenorrhea [39,40]. Expanding on this research, we explored interoception comparing patients with endometriosis and healthy controls, including a novel bladder interoception task, alongside established cardiac and gastric interoception tasks. Indeed, women with endometriosis suffer from pain mainly in the pelvic region which

topographically includes the bladder. Given this, urinary interoception is by far the easiest way to assess interoception in the pelvic area that, unfortunately, can be measured directly only through invasive procedures like intrapelvic ultrasound and cervical swabs [55,56]. Moreover, pelvic signals are mostly prominent in specific menstrual cycle phases and are measurable only via self-report questionnaires, which may be influenced by subjectivity. An important point of novelty of our study is the exploration of participants' perception of signals coming from the bladder in a non-invasive way, without exploiting medical equipment, making it accessible and easy to implement for future research. Our novel interoceptive urinary task (Urinary Water Load) positively correlated with the Two-step Water Load Test [35], while no correlation between these two and the Heartbeat Counting Task [28] was found. Thus, the higher the scores in the urinary interoception task, - indicating a better performance and a higher bladder interoceptive accuracy - the higher the scores in the gastric interoception task - implying a better gastric interoceptive accuracy.

These findings contribute to a broader debate within the literature on whether interoceptive abilities correlate across different body domains or remain modality-specific. For instance, Ferentzi et al. [57] proposed that interoceptive sensitivity is primarily a modality-specific feature, suggesting that individuals' accuracy in detecting internal signals is confined to specific bodily systems and does not extend to others (see [36,58] for other results supporting the modality-specific perspective). This perspective is supported by our observation that cardiac interoception, as measured by the Heartbeat Counting Task, did not correlate with either gastric or urinary interoception. In contrast, Whitehead and Drescher [59] argued for a more generalised tendency to be aware of or attend to visceral events, suggesting that individuals sensitive to one type of internal signal may also be sensitive to others (see [60–62] for other results supporting the generalised interoceptive ability perspective). This view aligns with our results showing a positive correlation between gastric and urinary interoception. Thus, our findings sit at a midway point in this debate, suggesting that interoceptive abilities may generalise within certain domains (e.g., visceral systems like the stomach and bladder) but remain distinct across others (e.g., cardiac versus non-cardiac domains). This may reflect the intrinsic difference between the visceral signals tackled in the three tasks. Indeed, heart signals at rest are less salient than those coming from the stomach and bladder. In other words, when not engaged in a specific task, individuals may not consistently prioritise signals originating from the cardiac region, whereas they tend to focus more on gastric and bladder signals due to their role in signalling urgent homeostatic and allostatic needs. Moreover, stomach and bladder signals share a neural pathway, as both gastric and bladder motility (i.e., the stomach contraction leading to stomach emptying and bladder contraction leading to urination) are controlled by an increased phasic activity of the parasympathetic pathways whereas the increase of the heart rate is guided by a higher phasic activity of the sympathetic pathway [63]. Another possibility is that the results of the gastric and urinary interoception accuracy tasks align in the same direction due to the intrinsic similarities in the nature of these tasks. Indeed, both tasks rely on visceral feedback during water-loading protocols, which may lead to overlapping cognitive and physiological processes. Additionally, recent studies show that tasks that share similar formats or instructions tend to show relationships, as participants may approach them with comparable strategies or mental frameworks [23].

Furthermore, we acknowledge as a limitation of this study solely using the Heartbeat counting task, suggesting that future studies should include an additional cardiac interoceptive accuracy task (e.g., Heartbeat discrimination task [64], as suggested by [65]) to better generalise this effect. Indeed, a debate in the literature exists pointing out that the perception of the heart signals during the Heartbeat Counting Task is influenced by several individual physiological and psychological factors [66] such as personal beliefs [67], heart rate variability [68], percentage of body fat [69] and systolic blood pressure [70]. Undoubtedly, these limitations of the HCT may have influenced our results. In particular, since individuals with chronic pain conditions are used to scanning their bodies for painful sensations, it is possible that patients with endometriosis compensate for reduced interoceptive accuracy by relying on knowledge of their heart rate, rather than directly perceiving their cardiac signals. This aligns with previous research showing that individuals with anxiety, who also tend to monitor bodily signals closely, may use cognitive strategies rather than true interoceptive perception in heartbeat detection

tasks [71]. Future studies could complement the HCT with additional cardiac interoceptive measures, such as the Heart-beat Discrimination Task, to better disentangle perceptual accuracy from cognitive estimation strategies.

Similarly, despite being correlated, the gastric and urinary interoceptive tasks assess two different constructs (i.e., participants' ability to perceive their stomach fullness in relation to the actual volume of water consumed the former, and participants' ability to discriminate between two different bladder stimuli the latter). Moreover, both tasks have limitations that need to be acknowledged. The principal drawback of the gastric water load task is that it relies on self-reported fullness, which is subjective and might be influenced by cognitive biases, mood, and prior experiences [35,72]. Thus, patients with endometriosis might report different sensations due to pain-related hypervigilance rather than their actual interoceptive accuracy. This could lead to overestimating or underestimating gastric sensations, making it difficult to tease apart true interoceptive accuracy from pain-related cognitive interference. Furthermore, chronic pelvic pain conditions like endometriosis are often associated with visceral hypersensitivity, which might amplify sensations of gastric discomfort. Patients could then experience gastric fullness at lower water volumes, not because of impaired interoception but due to visceral hypersensitivity. Moreover, since the test assesses fullness using water rather than solid food, its ecological validity is limited, making it less reflective of real-world eating behaviours. Lastly, individuals may interpret "maximum fullness" differently, leading to variability in results [35]. Concerning the urinary interoceptive task, its primary drawback lies in its time-consuming nature, an issue that should be addressed in future studies of urinary interoception. Additionally, future studies would benefit from integrating validated bladder and gastric symptom questionnaires, which could provide additional insights into the relationship between interoceptive accuracy and symptom burden.

Another limitation of this study is the absence of control tasks to assess whether participants manifest non-interoceptive perceptual impairments. This is particularly relevant for individuals suffering from chronic pain, as they may exhibit broader perceptual or attentional deficits that could influence their interoception task performances. Hence, considering this limitation, it may not be fully possible to draw firm conclusions regarding patients' interoceptive impairments, as differences may be due to non-interoceptive factors. Nevertheless, despite this limitation, our findings provide an important step forward in understanding interoception in this population, highlighting the need for future studies to incorporate control tasks to disentangle interoceptive and non-interoceptive contributions.

Concerning inclusion and exclusion criteria, patients with endometriosis adherences directly affecting the surface of the bladder were excluded based on clinical screening, to rule out the possibility that impaired perception of bladder signals was due to an organic issue caused by adhesions on the targeted organ (i.e., the bladder itself). Nevertheless, the exclusion of these patients limits the generalisability of our findings. Moreover, the use of hormonal contraception was assessed for the patients' group but not for the control group. This could constitute a limitation by causing the two tested samples to be less homogeneous. However, further analyses (see Supplementary Materials) revealed no significant differences in interoceptive accuracy between patients with different hormonal treatments (i.e., progestin, oestrogen, or no medication), suggesting that hormonal contraception did not substantially affect our main findings.

In addition, it is important to consider the potential influence of co-occurring mental health disorders on the observed differences in interoceptive accuracy between groups. Chronic health conditions like endometriosis are often associated with higher rates of anxiety and depression [73–75], which may also affect interoceptive processing [33]. Although our exclusion criteria included the presence of neurological or psychiatric conditions, we did not assess the possible presence of moderate mental health symptoms. Thus, we cannot rule out the possibility that patients with endometriosis suffered from subclinical depressive or anxious symptoms. Future studies should include questionnaires tackling mental health markers and, where possible, match groups on this variable to better disentangle the specific contributions of these factors to interoceptive accuracy.

Our results revealed that patients with endometriosis were significantly less accurate than control participants in discriminating between different signals coming from the stomach and bladder, but not from the heart.

Moreover, the results suggest that the lower performance of women with endometriosis in the urinary interoceptive task is not associated with previous endometriosis-related surgeries, nor by hormone-inhibiting drug intake. Conversely, interoceptive beliefs, namely the subjective tendency to focus on one's internal body signals, were comparable in endometriosis and healthy women as indexed by the MAIA-II questionnaire. Moreover, women with endometriosis reported higher bladder interoceptive belief scores compared to controls. Considering Suksasilp & Garfinkel's multi-dimensional interoceptive framework [23] and in line with recent literature [76], our data highlight the complexity of interoceptive processing in women with endometriosis. In particular, although when looking at MAIA-II questionnaire patients with endometriosis result comparable with controls in their subjective propensity to focus on internal body signals, they are ultimately less accurate than controls in discriminating urinary and gastric signals. Crucially, when looking specifically at urinary interoceptive beliefs (measured using the questions developed in this study), patients appear to focus more on bladder signals, experience greater interference with daily activities due to bladder stimuli, express greater concern about them, and experience more night and day time need to empty the bladder than controls. These findings shed light on a potential limitation of interoceptive questionnaires like the MAIA-II in assessing interoceptive beliefs, as they may fail to capture nuanced differences related to specific bodily domains. Thus, future studies could introduce more tailored tools capable of providing a precise evaluation of domain-specific interoceptive beliefs, particularly for patient populations with conditions like endometriosis that affect specific organs and body regions. Moreover, we speculate that patients may present a tendency to focus on their abdominal and pelvic areas, hyper-focusing on the pain in that area, rather than focusing on the physiological signals coming from that domain. This phenomenon is well exemplified by van Aken and colleagues [77], as part of the cognitive strategy of "catastrophising", frequently employed by patients with chronic pain, who perceive pain as highly threatening and show an excessive negative orientation toward pain, which makes it difficult for them to shift attention away from painful signals [78–80]. This "pain hypervigilance" frequently occurs in chronic pain patients, leading them to scan their bodies for pain sensations [77,81]. Future research should include a measure of this cognitive strategy to better explore this effect.

In addition to the urinary interoceptive index, this study considered a measure of consistency in perceiving the minimum interoceptive stimulus (i.e., void2 and void3). When exploring participants' consistency, the results show that both groups did not significantly differ in the amount of urine expelled during the minimum stimuli. In contrast, control participants were more accurate in perceiving the two different urinary stimuli (i.e., void1 vs. void2 and void3) than endometriosis patients. This suggests that endometriosis pain hypervigilance may selectively affect the ability to differentiate between interoceptive stimuli of varying intensities due to an increased attentional focus on urgency or discomfort. On the contrary, the perception of lower-intensity stimuli across repeated instances remains unaffected, as these may not trigger the same heightened salience.

Moreover, urinary consistency was not correlated to performance in the cardiac and gastric interoceptive tasks, suggesting that this measure captures a distinct facet of interoception. Conversely, the urinary interoceptive index and the gastric interoceptive index are correlated. Thus, it is possible that the discrimination between two stimulus intensities, rather than consistency in detecting the same stimulus intensity, may rely on shared mechanisms across gastric and urinary domains. These findings highlight the need for future research to consider interoception as a multidimensional construct, where different components may be differentially affected in clinical conditions such as chronic pain.

Our study also aimed to compare pain perception between patients with endometriosis and healthy women and the possible relationship between chronic pain and interoception. The interoceptive system, crucial for maintaining homeostatic control, integrates internal and external information to predict and manage bodily state changes [15,16,29,82,83] and it is tied to the *Salience* of an interoceptive stimulus, which depends upon its importance for bodily homeostasis [84]. Importantly, *the brain salience network* seems connected on several levels to the interoceptive cortex, which is crucial in predicting the interoceptive salience of a stimulus [85,86]. Thus, pain experience may derive from a predictive process involving top-down signals about the homeostatic significance of nociceptive inputs [29,87]. Another theory supporting the

link between the interoceptive cortex and the salience network relies on the concept of *Responsivity* [88], according to which the brain states are influenced by the dynamic integration of exteroceptive stimuli and interoceptive signals [29,88]. In line with this perspective, abnormal interpretations of interoceptive signal salience may lead to unpredictable responsivity in patients with chronic pain [88], ultimately leading to increased pain tolerance as a strategy for suppressing unnecessary inputs within an already overwhelmed system [21,29].

Furthermore, results show that women with endometriosis reported experiencing higher pelvic pain, pain during menstruation, sexual intercourse, and defecation than controls. The moderating effect of the group on the relationship between interoception and chronic pain is not large enough to draw firm conclusions. In order to better understand the relationship between interoception and pain in endometriosis, future studies should include a measure assessing objective pain, which was lacking in the present study. Indeed, electrophysiological pain markers (e.g., EEG [89,90]; EMG [91]) and pain threshold measures (e.g., electrical stimulation [92]; pressure stimulation [93]; thermal stimulation [94]) should complement the use of questionnaires for assessing pain in future studies.

Regarding self-report questionnaires, results from the Depersonalisation subscale of the Body Uneasiness Test questionnaire [50] suggest that women with endometriosis have a greater tendency to detach and estrange themselves from their bodies than healthy controls. This aligns with literature linking dissociation episodes to the development or increase of chronic pain [95] and reporting higher dissociative experiences in chronic pain [96] or chronic pelvic pain [97] patients than in controls. These dissociation episodes may represent a coping strategy in response to the continuous or repetitive pain in these conditions [98].

This study introduces a non-invasive tool for assessing bladder interoception, which is topographically close to the uterus. On the one hand, identifying interoceptive markers in women with endometriosis may represent a potential aid for diagnosis. However, further studies are needed to corroborate these findings using different methodological approaches. On the other hand, the new urinary interoceptive task paves the way for the development of non-pharmacological therapeutic approaches aimed at improving patients' interoceptive abilities, such as the biofeedback approach relying on cardiac interoception [99]. Protocols combining biofeedback and interoceptive training could be designed in the future to improve interoceptive accuracy in women with endometriosis and shift their focus from pain to the perception of visceral signals. These proposed interventions remain speculative as the research on urinary interoceptive abilities is still in the early stages.

## Supporting information

**S1 File. Supplementary materials.** File including all supplementary figures, tables, and analyses.
(PDF)

## Acknowledgments

The support of the Paris Institute for Advanced Study to Salvatore Maria Aglioti is gratefully acknowledged. We thank *Alice O.D.V. Italian Association for Endometriosis* members for their help in the recruitment, Andrea Ciccarone for his support in refining our reasoning, and women from all over Italy who volunteered for the study.

## Author contributions

**Conceptualization:** Chiara Cantoni, Sofia Ciccarone, Maria Grazia Porpora, Salvatore Maria Aglioti.

**Data curation:** Chiara Cantoni, Sofia Ciccarone.

**Formal analysis:** Chiara Cantoni, Sofia Ciccarone.

**Funding acquisition:** Salvatore Maria Aglioti.

**Investigation:** Chiara Cantoni, Sofia Ciccarone.

**Methodology:** Chiara Cantoni, Sofia Ciccarone.

**Supervision:** Maria Grazia Porpora, Salvatore Maria Aglioti.

**Visualization:** Chiara Cantoni, Sofia Ciccarone.

**Writing – original draft:** Chiara Cantoni.

**Writing – review & editing:** Sofia Ciccarone, Maria Grazia Porpora, Salvatore Maria Aglioti.

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
