## [Decision Letter · Decision Letter 0]

7 Jan 2025

PONE-D-24-50496Impaired gastric and urinary but preserved cardiac interoception in women with endometriosisPLOS ONE

Dear Dr. Cantoni,

Thank you for submitting your manuscript to PLOS ONE. The reviewers were positive about many aspects of your manuscript but have also raised a number of concerns and have provided several significant recommendations for revisions. I therefore, we invite you to submit a revised version of the manuscript that addresses the points raised during the review process.

We look forward to receiving your revised manuscript.

Kind regards,

Prof. Jane Elizabeth Aspell, PhD

Academic Editor

PLOS ONE

Journal Requirements:

[This work was supported by a European Research Council (ERC, https://erc.europa.eu/homepage ) Advanced Grant 2017, Embodied Honesty in real world and digital interactions (eHONESTY, Prot. 789058) (to S.M.A.) and Ministero dell'Università e della Ricerca (PRIN n. 20229JPNT7, https://prin.mur.gov.it/ ) (to S.M.A.)].

3. Thank you for uploading your study's underlying data set. Unfortunately, the repository you have noted in your Data Availability statement does not qualify as an acceptable data repository according to PLOS's standards.

Reviewers' comments:

Reviewer's Responses to Questions

**Comments to the Author**

1. Is the manuscript technically sound, and do the data support the conclusions?

Reviewer #1: Partly

Reviewer #2: Yes

2. Has the statistical analysis been performed appropriately and rigorously? 

Reviewer #1: No

Reviewer #2: Yes

3. Have the authors made all data underlying the findings in their manuscript fully available?

Reviewer #1: No

Reviewer #2: No

4. Is the manuscript presented in an intelligible fashion and written in standard English?

Reviewer #1: Yes

Reviewer #2: Yes

5. Review Comments to the Author

Reviewer #1: Thank you for inviting me to review this paper. I love the focus – both in terms of an under researched condition, as well as the development of novel tests and the focus on signals beyond cardiac interoceptive accuracy. I really wanted to like this, but I have major concerns about the paper in its present form. These are detailed below, and I am on the fence between recommending rejection vs. major revision. On balance, I think that the authors should have the opportunity to improve their paper because this is a difficult and novel question, but there is a fair bit of work to be done.

Major:

1. Introduction - I think we should be careful to say that patients ‘misperceive’ signals given that the tasks used may have reporting biases (e.g., HCT, Water-load) and some studies in chronic pain show increased interoceptive accuracy (e.g., Todd et al., 2024; using the PAT).

2. Introduction - The 2015 Garfinkel model is quite out of date now. Perhaps use the updated paper from those authors detailed below:

a. Suksasilp, C., & Garfinkel, S. N. (2022). Towards a comprehensive assessment of interoception in a multi-dimensional framework. Biological Psychology, 168, 108262.

3. Introduction – we really shouldn’t be using the term interoceptive sensibility anymore given a wealth of evidence that different measures tap different processes (see references below). Please be more specific throughout the manuscript. I would also not rely on evidence from the HCT given controversy about this task. I would like to see more nuance in the introduction.

a. Desmedt, O., Luminet, O., Walentynowicz, M., & Corneille, O. (2023). The new measures of interoceptive accuracy: a systematic review and assessment. Neuroscience & Biobehavioral Reviews, 105388.

b. Desmedt, O., Heeren, A., Corneille, O., & Luminet, O. (2022). What do measures of self-report interoception measure? Insights from a systematic review, latent factor analysis, and network approach. Biological Psychology, 169, 108289.

4. Methods – 30 participants per group is tiny. Although a power analysis is included, there is no justification for the expected effect size of η2=0.14 which is VERY large. Perhaps this should be framed as a pilot study as the sample sizes are really too low to make firm conclusions. This will mean tempering of the quite bold conclusions in the Discussion as what is really needed is a well-powered replication of this effect, certainly not the development of interventions on the basis of this small study.

5. Methods – please provide a justification for this exclusion criteria “presence of endometriosis adhesions in the bladder area” – wouldn’t these be the people we would want to include?

6. Methods - “chronic intake of drugs (except for those taken to treat endometriosis)” – please could you provide these details. For example, are more women on hormonal contraception in your endometriosis group than control group? Do you have differences in mental health between the groups? In the Discussion there is discussion of catastrophising and symptoms that very much seem to link to anxiety.

7. Methods – please could you include a justification for why no control tasks were included as part of the procedure to demonstrate no general perceptual issues that are non-interoceptive. This is a major limitation that needs to be acknowledged. For individuals who have chronic pain, this is particularly important.

8. Methods – I am not a fan of the HCT given the potential for reporting bias, nor the water load test given that the two-stage procedure may still be influenced by beliefs about stomach capacity and feelings of fullness. I am not entirely sure that the calculation of the urinary task actually provides the metric the authors are after. If you simply are looking for the consistency in minimum, would you not simply take the absolute difference between the two divided by total capacity. I am not sure why you have taken an average (e.g., divided by two). I would think this should be ABS(void 2 – void 3)/void 1.

9. Methods – out of interest, were void 2 or void 3 consistently greater or larger than each other in either group?

10. Methods – could you provide a rationale for why you didn’t include a validated bladder symptom or gastric symptom questionnaire?

11. Data analysis – the sample is far too small to run a factor analysis unless this has been validated elsewhere and should not be used. I would suggest looking at individual items and correcting for comparison.

12. Results – as your correlation between tasks is driven by the healthy controls, could you please colour code groups in figure 3 and add the results from the supplement in the figure legend. Also, in figure legends please include the direction of scores – e.g., high score = better perception.

13. Results – I think that given so many analyses, I would not discuss marginally significant results. As you have a lot of questionnaires/analyses, where these are answering the same question, I think it would be good to include a correction for all of these comparisons. Not being too strict, but maybe p=.025 should be the alpha here as there are many comparisons. At a minimum, you should detail which survive correction for multiple comparisons where the same family of tests answer the same question (e.g., if looking at patients vs. controls on interoceptive accuracy tasks, you should correct for the three comparisons).

14. Figures – have come out a bit blurry. Could you try making it easier to see the words in the figures.

15. Discussion – when discussing the results of the gastric and urinary task relationship, it would be nice to also acknowledge that tasks that share similar formats/demands are often related and discuss this with a view to studies that don’t often report relationships across domains. See reference below for example.

a. Whitehead, W. E., & Drescher, V. M. (1980). Perception of gastric contractions and self‐control of gastric motility. Psychophysiology, 17(6), 552-558.

b. Ferentzi, E., Bogdány, T., Szabolcs, Z., Csala, B., Horváth, Á., & Köteles, F. (2018). Multichannel investigation of interoception: Sensitivity is not a generalizable feature. Frontiers in human neuroscience, 12, 223.

16. Discussion – not sure if you should recommend the Whitehead et al., (1977) procedure. There is a wealth of data on the issues with this task. See the review below.

a. Brener, J., & Ring, C. (2016). Towards a psychophysics of interoceptive processes: the measurement of heartbeat detection. Philosophical Transactions of the Royal Society B: Biological Sciences, 371(1708), 20160015.

17. Discussion – somewhere you should outline the issues with the other two tasks not just the HCT, including that they aren’t exactly measuring the same thing despite being correlated (fullness vs consistency) as well as discuss their limitations.

18. Discussion – I think that in discussion of the HCT, one should be careful to describe not just general limitations, but how this would have influenced YOUR results. E.g., perhaps patients compensate for poor perception using heart rate knowledge which they may be expected to have more of, given they may monitor their bodily signals more. Some evidence to this effect here for anxiety:

a. Biotti, F., Barker, M., Carr, L., Pickard, H., Brewer, R., & Murphy, J. The effects of the SARS-CoV-2 pandemic on self-reported. PloS one.

19. Discussion – relatedly, as chronic health conditions are often associated with poor mental health, one should also acknowledge that differences may not be driven by the condition, but may be driven by other co-occurring conditions as presumably groups were not matched for this.

20. Discussion – I do not agree that these results suggest a distinction between objective vs. self-report, I think you should be careful given there are self-reported differences when you look at bladder symptoms, specifically. As such, I would acknowledge here the possibility that general measures may not capture specific differences that relate to a single bodily domain.

21. Discussion – the paragraph related to neural underpinnings is unnecessary and can be removed.

22. Discussion – I would not discuss the result that was not significant in as much detail.

Minor:

1. Abstract – should be cardiac interoception (or interoceptive accuracy) tasks.

1. Introduction - Is the term interstitial cystitis still in use? I thought it was now called chronic urinary tract infections.

2. Introduction – please use consistent terminology throughout (e.g., domains, different bodily districts etc).

Reviewer #2: The study compared interoceptive accuracy and interoceptive beliefs in women with endometriosis and healthy controls across cardiac, gastric, and urinary domains. Women with endometriosis showed reduced accuracy in gastric and urinary interoception but not cardiac interoception. Despite this, they reported heightened bladder interoceptive beliefs, characterised by increased attention and concern about bladder signals. Group marginally moderated the effect of chronic pain on urinary interoception, suggesting that heightened pain sensitivity may impair interoceptive accuracy. The novel Urinary Interoceptive Task correlated with gastric but not cardiac interoceptive accuracy, highlighting shared neural pathways for gastric and bladder signals.

This study highlights dissociations between objective interoceptive accuracy and subjective interoceptive beliefs in women with endometriosis, suggesting that interoceptive impairments may result from pain hypersensitivity and hypervigilance. The findings provide valuable insights for developing non-invasive diagnostic tools and interoceptive training therapies for chronic pain management. This study addresses a relevant topic in interoception and employed a well-designed method. The results are informative and important in further elucidating the relationship between chronic pain and interoception. The manuscript offers an interesting read and would make a meaningful contribution to the interoception and pain literature following revision and clarification of the following details:

Introduction

The terminology around interoceptive dimensions would benefit from updating to match the more recent multidimensional framework by Suksasilp & Garfinkel (2022), Towards a comprehensive assessment of interoception in a multi-dimensional framework, and change terms such as interoceptive sensibility to interoceptive beliefs, and interoceptive awareness to interoceptive insight, accordingly.

Method

POWER analysis: More detail is required regarding the test used. Please clarify which test family, statistical test, type of power analysis, and other relevant input parameters (e.g. number of tails) were entered.

Water load test:

How long apart were the two drinking phases; i.e., did drinking phase 2 follow phase 1 immediately?

State the meaning of the index regarding interoceptive accuracy (higher index, higher accuracy?)

Urinary interoception task:

On page 7, lines 172–174, you mention the use of a VAS for participants to rate their bladder sensations. Could you clarify in the manuscript which specific rating on the scale (0–10) was defined as the 'minimum urinary stimulus'?

To enhance the reliability and validity of the findings, please clarify whether the same subjective rating of the 'minimum urinary stimulus' was consistently applied within participants across the void2ml and void3ml conditions. For example, if a participant indicated '4' as the minimum in the void2ml condition, was the same rating ('4') considered as the minimum in the void3ml condition?

If consistency was not maintained, I recommend noting this as a limitation in the manuscript, given the importance of this measure in calculating the Urinary Interoceptive Index. Specifically, discuss how variability in subjective thresholds and baseline urinary sensations might have influenced condition comparability (e.g., void2ml vs. void3ml within participants), introduced noise into the data, or affected the interpretation of findings.

Line 183: Replace "her" with "their" to ensure inclusive language.

6. PLOS authors have the option to publish the peer review history of their article (what does this mean? ). If published, this will include your full peer review and any attached files.

**Do you want your identity to be public for this peer review?** For information about this choice, including consent withdrawal, please see our Privacy Policy .

Reviewer #1: No

Reviewer #2: **Yes: ** Gaby Pfeifer

---

## [Author Response · Author response to Decision Letter 0]

7 Feb 2025

We thank the Editor for giving us the opportunity to reply to the reviewers' comments and to improve our manuscript. Please see the file "Response to Reviewers" for an extended version of the response to the reviewers that follows.

We thank the Editor for the suggestion. As previously stated in the first submission “The datasets generated and analysed during the current study and supplementary materials, and the components of the research methodology needed to reproduce the reported procedure(s) and analyses are publicly available on the Open Science Framework repository (DOI: 10.17605/OSF.IO/H7V5Y)”. We decided to share our data on the Open Science Framework repository as it is among the suggested repositories of PLOS ONE. In order to comply with the Editor’s request, we also shared our laboratory protocol in protocols.io at the following DOI: dx.doi.org/10.17504/protocols.io.yxmvm9m7nl3p/v1

Journal Requirements:

We edited the manuscript in order to meet PLOS ONE's formatting requirements.

[This work was supported by a European Research Council (ERC, https://erc.europa.eu/homepage) Advanced Grant 2017, Embodied Honesty in real world and digital interactions (eHONESTY, Prot. 789058) (to S.M.A.) and Ministero dell'Università e della Ricerca (PRIN n. 20229JPNT7, https://prin.mur.gov.it/) (to S.M.A.)].

We included the Role of Funder statement in our cover letter as required. The cover letter now reads as follows: “This work was supported by a European Research Council (ERC, https://erc.europa.eu/homepage) Advanced Grant 2017, Embodied Honesty in real world and digital interactions (eHONESTY, Prot. 789058) (to S.M.A.) and Ministero dell'Università e della Ricerca (PRIN n. 20229JPNT7, https://prin.mur.gov.it/) (to S.M.A.). The funders had no role in study design, data collection and analysis, decision to publish, or preparation of the manuscript.”

3. Thank you for uploading your study's underlying data set. Unfortunately, the repository you have noted in your Data Availability statement does not qualify as an acceptable data repository according to PLOS's standards.

Although we already published our data in the Open Science Framework (included in your list of recommended repositories) at the following link: DOI: 10.17605/OSF.IO/H7V5Y, we also uploaded datasets and codes in Figshare at the following DOI: 10.6084/m9.figshare.28369424

5. Review Comments to the Author

Reviewer #1: Thank you for inviting me to review this paper. I love the focus – both in terms of an under researched condition, as well as the development of novel tests and the focus on signals beyond cardiac interoceptive accuracy. I really wanted to like this, but I have major concerns about the paper in its present form. These are detailed below, and I am on the fence between recommending rejection vs. major revision. On balance, I think that the authors should have the opportunity to improve their paper because this is a difficult and novel question, but there is a fair bit of work to be done.

Major:

1. Introduction - I think we should be careful to say that patients ‘misperceive’ signals given that the tasks used may have reporting biases (e.g., HCT, Water-load) and some studies in chronic pain show increased interoceptive accuracy (e.g., Todd et al., 2024; using the PAT).

We thank the reviewer for their insightful suggestion. We agree that the term “misperceive” is not fully exhaustive for all the cases regarding chronic pain patients, even though the majority of the studies indicate a reduced ability to perceive internal bodily signals in this population. To address this, we have revised the text in the Introduction to replace the term "misperceive" and instead described interoceptive alterations or differences in interoceptive abilities between chronic pain patients and healthy controls. Moreover, we added the suggested reference (i.e. Todd et al., 2024), whose results are in contrast with the one we previously cited. We hope that this adjustment ensures that the language better reflects the complexity of interoceptive processes and the potential impact of task-specific characteristics. The sentence on line 37, page 3, now reads as follows: “Generally, individuals suffering from chronic pain and visceral hypersensitivity (i.e., increased sensitivity to pain originating from the internal organs) typically exhibit altered interoception, i.e. they perceive the physiological state of their body differently if compared to healthy controls (i.e. lower interoceptive accuracy [13–21] or higher interoceptive accuracy [Todd et al., 2024].”

2. Introduction - The 2015 Garfinkel model is quite out of date now. Perhaps use the updated paper from those authors detailed below:

a. Suksasilp, C., & Garfinkel, S. N. (2022). Towards a comprehensive assessment of interoception in a multi-dimensional framework. Biological Psychology, 168, 108262.

We thank the reviewer for their insightful suggestion. Initially, we decided to apply the Garfinkel et al. (2015) model to our project because, at the time of its conceptualisation, the Suksasilp and Garfinkel (2022) article had not yet been published. However, we agree on updating the interoception model to encompass all aspects currently considered in interoception field research. Therefore, we have revised the literature cited in the introduction, methods, and discussion sections of the manuscript, updating the terminology in line with the paper by Suksasilp and Garfinkel (2022).

3. Introduction – we really shouldn’t be using the term interoceptive sensibility anymore given a wealth of evidence that different measures tap different processes (see references below). Please be more specific throughout the manuscript. I would also not rely on evidence from the HCT given the controversy about this task. I would like to see more nuance in the introduction.

a. Desmedt, O., Luminet, O., Walentynowicz, M., & Corneille, O. (2023). The new measures of interoceptive accuracy: a systematic review and assessment. Neuroscience & Biobehavioral Reviews, 105388.

b. Desmedt, O., Heeren, A., Corneille, O., & Luminet, O. (2022). What do measures of self-report interoception measure? Insights from a systematic review, latent factor analysis, and network approach. Biological Psychology, 169, 108289.

We thank the reviewer for this comment and the suggested references. We appreciate the opportunity to refine the manuscript in light of recent evidence. First of all, we agree that the term "interoceptive sensibility" may no longer fully capture the complexity of processes measured by self-report tools. To address this issue, we revised the manuscript using more specific and up-to-date terminology, such as "self-reported interoceptive beliefs" or "subjective interoceptive beliefs" as in Suksasilp & Garfinkel (2022). We also explicitly addressed the evolving understanding of interoceptive constructs in the introduction by adding the following sentence on line 51, page 3: “[...] (for an exhaustive description of all the different dimensions of interoception please refer to [22]) [...]”. Furthermore, we acknowledged that results based on questionnaires assessing subjective interoceptive beliefs should be interpreted with caution, due to the concerns on interoceptive questionnaires recently raised by Desmedt and colleagues (2022). For this reason, we edited the introduction at line 58, page 4, which now reads as follows: “[...] Recent studies using subjective assessments have demonstrated that interoceptive self-regulation mediates the relationship between pain and depression in endometriosis patients, underscoring the crucial role of interoception in this condition [32]. Nevertheless, Desmedt and colleagues (2022) suggest that uncertainty remains about whether current "interoceptive sensibility" and "self-report interoceptive scales" measures really assess a common construct. [...]”

Secondly, for what concerns the Heartbeat counting task (HCT), we acknowledge the ongoing debate about the validity of this task as a measure of interoceptive accuracy. To address this, we included a brief discussion in the introduction about the limitations of the HCT, citing relevant literature (e.g., Desmedt et al., 2023). Now the introduction on line 66, page 4 reads as follows: “[...] Patients and healthy controls performed the Heartbeat Counting Task to assess the cardiac domain [27], the Two-step Water Load Test for the gastric domain [33] and a novel version of a urinary interoceptive task designed to test bladder interoception. However, it is important to acknowledge the limitations of the Heartbeat Counting Task (HCT, [27]), which has been widely used to measure interoceptive cardiac accuracy. Recent evidence has raised concerns about the validity of the HCT, suggesting that it may not purely assess interoceptive accuracy but could also be influenced by other factors such as general cognitive abilities, beliefs about heart rate, and knowledge of cardiac rhythms. For instance, Desmedt et al. (2023) highlighted that the HCT results may reflect participants’ expectations or their ability to estimate time rather than their capacity to accurately perceive cardiac signals. These limitations underscore the importance of complementing this cardiac task with additional interoceptive tasks that assess other domains and provide a more comprehensive assessment of interoceptive abilities.[...]” We hope these revisions address the reviewer’s concerns and improve the clarity and rigour of our manuscript.

4. Methods – 30 participants per group is tiny. Although a power analysis is included, there is no justification for the expected effect size of η2=0.14 which is VERY large. Perhaps this should be framed as a pilot study as the sample sizes are really too low to make firm conclusions. This will mean tempering of the quite bold conclusions in the Discussion as what is really needed is a well-powered replication of this effect, certainly not the development of interventions on the basis of this small study.

We thank the reviewer for this comment. As mentioned by the reviewer, a power analysis was performed based on the power analysis and effect size indicated by Di Lernia et al., 2020 (f = 0.4, α err prob. = 0.05, power = 0.80) and Salamone et al., 2021 ( f = 0.40, α level of p = 0.05, and a power of 0.80), both including interoception tasks and patients populations (Di Lernia et al., 2020 recruited chronic pain patients specifically). Moreover, studies regarding interoceptive abilities in chronic pain conditions usually recruit a comparable number of patients and controls: Solcà et al., 2020 (24 chronic pain patients vs. 24 HC); Martinez et al., 2018 (14 chronic pain patients vs. 13 HC); d'Alcalà, Webster & Esteves, 2015 (22 chronic pain patients vs. 37 HC); Borg et al., 2018 (21 fibromyalgia patients vs. 21 HC); Shizuma et al., 2021 (38 chronic pain vs. 35 HC); Valenzuela-Moguillansky, et al., 2017 (30 fibromyalgia patients vs. 29 HC). Thus, while we appreciate the reviewer’s suggestion to frame the study as a pilot, we would like to emphasize that the sample size in our study is consistent with that of several studies in the existing literature, which were not framed as pilot studies.

Nevertheless, we tried to soften the conclusions the reviewer is referring to, underlying the necessity of further studies to corroborate our results and suggesting the development of interventions as a further step forward. The edited discussion paragraph concerning the conclusions, on line 483, page 19, now reads as follows: “[...] This study introduces a non-invasive tool for assessing bladder interoception, which is topographically close to the uterus. On the one hand, identifying interoceptive markers in women with endometriosis may represent a potential aid for diagnosis. However, further studies are needed to corroborate these findings using different methodological approaches. On the other hand, the new urinary interoceptive task paves the way for the development of non-pharmacological therapeutic approaches aimed at improving patients' interoceptive abilities, such as the biofeedback approach relying on cardiac interoception [81]. Protocols combining biofeedback and interoceptive training could be designed in the future to improve interoceptive accuracy in women with endometriosis and shift their focus from pain to the perception of visceral signals. These proposed interventions remain speculative as the research on urinary interoceptive abilities is still in the early stages.”

5. Methods – please provide a justification for this exclusion criteria “presence of endometriosis adhesions in the bladder area” – wouldn’t these be the people we would want to include?

We thank the reviewer for the request for clarification. Indeed, the exclusion criterion “presence of endometriosis adhesions in the bladder area” seems to suggest that we excluded participants with adhesions in the pelvic region, not only specifically located on the bladder itself (e.g., adhesions on the ovaries, Fallopian tubes, pelvic peritoneum, uterosacral ligaments, or the external surface of the uterus). Nevertheless, we did not include in our sample women with endometriosis directly affecting the surface of the bladder, in order to rule out the possibility that impaired perception of bladder signals was due to an organic issue caused by adhesions on the organ being tested (i.e. the bladder itself). To avoid this confounding factor, we excluded women with bladder-specific adhesions and instead included those with adhesions in other pelvic areas. To further clarify this point, we revised the description of the exclusion criterion to specifically refer to “adhesions on the bladder” rather than the broader term “adhesions in the bladder area.” We hope this adjustment clarifies our approach.

6. Methods - “chronic intake of drugs (except for those taken to treat endometriosis)” – please could you provide these

---

## [Decision Letter · Decision Letter 1]

4 Mar 2025

PONE-D-24-50496R1Impaired gastric and urinary but preserved cardiac interoception in women with endometriosisPLOS ONE

Dear Dr. Cantoni,

Thank you for submitting your manuscript to PLOS ONE. I have received two expert reviews and while one reviewer is satisfied with the current manuscript, the other has some remaining concerns which need to be addressed. Therefore, we invite you to submit a revised version of the manuscript that addresses the points raised during the review process.

We look forward to receiving your revised manuscript.

Kind regards,

Jane Elizabeth Aspell, PhD

Academic Editor

PLOS ONE

Journal Requirements:

Reviewers' comments:

Reviewer's Responses to Questions

**Comments to the Author**

1. If the authors have adequately addressed your comments raised in a previous round of review and you feel that this manuscript is now acceptable for publication, you may indicate that here to bypass the “Comments to the Author” section, enter your conflict of interest statement in the “Confidential to Editor” section, and submit your "Accept" recommendation.

Reviewer #1: (No Response)

Reviewer #2: All comments have been addressed

2. Is the manuscript technically sound, and do the data support the conclusions?

Reviewer #1: Partly

Reviewer #2: Yes

3. Has the statistical analysis been performed appropriately and rigorously? 

Reviewer #1: No

Reviewer #2: Yes

4. Have the authors made all data underlying the findings in their manuscript fully available?

Reviewer #1: Yes

Reviewer #2: Yes

5. Is the manuscript presented in an intelligible fashion and written in standard English?

Reviewer #1: Yes

Reviewer #2: Yes

6. Review Comments to the Author

Reviewer #1: Thank you to the authors for their efforts in responding to comments. This is mostly really well achieved. I only have a few remaining concerns.

1. Bladder endometriosis exclusion and the issue of hormonal contraception – I would like to see those added as a limitation alongside the issue of mild/moderate mental health symptoms please.

2. I would like to see some discussion of the issues with the waterload task, alongside the issues with the HCT.

3. I’m afraid the lack of a control task is unjustifiable – without these, you cannot know that differences between groups are due to interoceptive abilities. The fact that other studies in patients are also of poor quality is not a defence. The limitation should be revised to point out that no clear conclusions regarding interoceptive ability can be made without the use of control tasks as differences may be due to non-interoceptive factors. This point should be made clearly and explicitly, at the moment this is framed as a future direction rather than a critical limitation.

4. Analysis of the urinary task. In the example provided, both participants show perfect consistency (their minimum stimulus is the same both times, suggesting that when asked to perceive the same amount, they are consistent (i.e. highly interoceptive). The discrimination between the minimum and maximum does not provide this as it would be influenced by bladder capacity etc., I appreciate the authors adding this in the supplement (where no differences were found), but I think this should really be included in the main text, in relation to relationships between tasks as well, and in the discussion the differences between the metrics should be clearly articulated with some discussion of what this means for interpretations.

Reviewer #2: I have read the rebuttal and revised manuscript and am satisfied with the authors' changes.

Key concerns, including updating interoceptive terminology to align with the multidimensional framework by Suksasilp & Garfinkel (2022), clarifying the power analysis details, and providing further methodological and analytical clarifications regarding the urinary interoception task, have been adequately addressed.

The authors have also improved transparency in their reporting of the water load test, ensured inclusive language, and discussed potential limitations where appropriate.

The manuscript is engaging, and the results provide meaningful insights into interoceptive accuracy in patients with endometriosis across different bodily axes, contributing to interoception and chronic pain research.

7. PLOS authors have the option to publish the peer review history of their article (what does this mean? ). If published, this will include your full peer review and any attached files.

**Do you want your identity to be public for this peer review?** For information about this choice, including consent withdrawal, please see our Privacy Policy .

Reviewer #1: No

Reviewer #2: No

---

## [Author Response · Author response to Decision Letter 1]

12 Mar 2025

We thank the Editor for giving us the chance to further improve our manuscript by addressing reviewers’ comments. Please find below a point-by-point reply to the remarks of the reviewers with changes to the original manuscript made after the first rebuttal highlighted in yellow and the changes made after the second rebuttal highlighted in light blue. We hope that you and the reviewers are happy with the revised manuscript and that it can be accepted for publication.

Review Comments to the Author

Reviewer #1: Thank you to the authors for their efforts in responding to comments. This is mostly really well achieved. I only have a few remaining concerns.

1. Bladder endometriosis exclusion and the issue of hormonal contraception – I would like to see those added as a limitation alongside the issue of mild/moderate mental health symptoms, please.

We thank the reviewer for allowing us to further improve our manuscript. We edited the limitation section of the discussion by following the reviewer’s suggestion. The paragraph at line 432, page 17, now reads as follows: “Concerning inclusion and exclusion criteria, patients with endometriosis adherences directly affecting the surface of the bladder were excluded based on clinical screening, to rule out the possibility that impaired perception of bladder signals was due to an organic issue caused by adhesions on the targeted organ (i.e. the bladder itself). Nevertheless, the exclusion of these patients limits the generalisability of our findings. Moreover, the use of hormonal contraception was assessed for the patients’ group but not for the control group. This could constitute a limitation by causing the two tested samples to be less homogeneous. However, further analyses (see Supplementary Materials) revealed no significant differences in interoceptive accuracy between patients with different hormonal treatments (i.e., progestin, oestrogen, or no medication), suggesting that hormonal contraception did not substantially affect our main findings.

In addition, it is important to consider the potential influence of co-occurring mental health disorders, on the observed differences in interoceptive accuracy between groups. Chronic health conditions like endometriosis are often associated with higher rates of anxiety and depression [73-75], which may also affect interoceptive processing [33]. Although our exclusion criteria included the presence of neurological or psychiatric conditions, we did not assess the possible presence of moderate mental health symptoms. Thus, we cannot rule out the possibility that patients with endometriosis suffered from subclinical depressive or anxious symptoms. Future studies should include questionnaires tackling mental health markers and, where possible, match groups on this variable to better disentangle the specific contributions of these factors to interoceptive accuracy.”

We hope that adding this paragraph we complied with the reviewer’s request.

2. I would like to see some discussion of the issues with the waterload task, alongside the issues with the HCT.

We thank the reviewer for requesting further details on the limitations of the Waterload task. Following the reviewer’s previous comments, we already included a paragraph concerning the general issues of the WTL in the discussion. Therefore, we assumed that the reviewer would like us to add further details concerning how these limitations could affect our results specifically. Thus, we edited the discussion paragraph on line 410, page 17, as follows: “[…] Moreover, both tasks have limitations that need to be acknowledged. The principal drawback of the gastric water load task is that it relies on self-reported fullness, which is subjective and might be influenced by cognitive biases, mood, and prior experiences [35,72]. Thus, patients with endometriosis might report different sensations due to pain-related hypervigilance rather than their actual interoceptive accuracy. This could lead to overestimating or underestimating gastric sensations, making it difficult to tease apart true interoceptive accuracy from pain-related cognitive interference. Furthermore, chronic pelvic pain conditions like endometriosis are often associated with visceral hypersensitivity, which might amplify sensations of gastric discomfort. Patients could then experience gastric fullness at lower water volumes, not because of impaired interoception but due to visceral hypersensitivity. Moreover, since the test assesses fullness using water rather than solid food, its ecological validity is limited, making it less reflective of real-world eating behaviours. Lastly, individuals may interpret ‘maximum fullness’ differently, leading to variability in results [35]. […]”

We hope that with this integration, we addressed the reviewer’s concerns.

3. I’m afraid the lack of a control task is unjustifiable – without these, you cannot know that differences between groups are due to interoceptive abilities. The fact that other studies in patients are also of poor quality is not a defence. The limitation should be revised to point out that no clear conclusions regarding interoceptive ability can be made without the use of control tasks as differences may be due to non-interoceptive factors. This point should be made clearly and explicitly, at the moment this is framed as a future direction rather than a critical limitation.

We thank the reviewer again for pointing out this limitation. We rephrased the paragraph to make it clear that the lack of a control task constitutes a critical limitation. The paragraph at line 427, page 17, now reads as follows: “Another limitation of this study is the absence of control tasks to assess whether participants manifest non-interoceptive perceptual impairments. This is particularly relevant for individuals suffering from chronic pain, as they may exhibit broader perceptual or attentional deficits that could influence their interoception task performances. Hence, future studies should include control tasks that help isolate interoceptive-specific impairments from more generalised perceptual deficits. Hence, considering this limitation, it may not be fully possible to draw firm conclusions regarding patients’ interoceptive impairments, as differences may be due to non-interoceptive factors.” Nevertheless, despite this limitation, our findings provide an important step forward in understanding interoception in this population, highlighting the need for future studies to incorporate control tasks to disentangle interoceptive and non-interoceptive contributions.

4. Analysis of the urinary task. In the example provided, both participants show perfect consistency (their minimum stimulus is the same both times, suggesting that when asked to perceive the same amount, they are consistent (i.e. highly interoceptive). The discrimination between the minimum and maximum does not provide this as it would be influenced by bladder capacity etc., I appreciate the authors adding this in the supplement (where no differences were found), but I think this should really be included in the main text, in relation to relationships between tasks as well, and in the discussion the differences between the metrics should be clearly articulated with some discussion of what this means for interpretations.

We thank the reviewer for highlighting this perspective on the analysis of the urinary task. We want to clarify that we conceptualised the Urinary Interoceptive Index as the result of the ratio between the mean of the two minimum stimuli and the maximum stimulus for different reasons. First, it reflects the idea of the gastric Water load by Van Dyck et al. (2016), where there are two different stimuli to reach and compare (i.e., satiety and fullness for the gastric water load and minimum and maximum stimuli for the urinary water load). Second, we do not believe that the index would be strongly influenced by bladder capacity: for the way we conceptualised the index, a participant with a small bladder will reach both the maximum and the minimum stimuli at lower millilitres of urine if compared to a participant with a bigger bladder, but the ratio will stay the same. Thus, participants are accurate in detecting the difference between the two urinary stimuli, their urinary index will result higher than participants who are not as accurate, regardless of the size of their bladder. Last, we think that comparing the minimum vs. maximum urinary stimuli is more ecological than just comparing the two minimum stimuli, as people in everyday life indeed perceive the urinary stimuli at different intensities, reflecting different urgencies.

Nevertheless, we think that, as suggested by the reviewer, adding the measure reflecting participants’ consistency in detecting their minimum stimuli to the results and the discussion would improve the manuscript from its current state.

As suggested by the reviewer, we moved the analysis concerning the comparison between the two minimum stimuli from the supplementary materials to the main text. Data Analysis and Results sections now read as follows:

Data analysis, “Relationship between the Interoceptive Tasks” paragraph, line 216, page 10:

“We also wanted to explore if participants’ consistency in discriminating the two minimum urinary stimuli correlated with their ability to detect gastric and cardiac stimuli, measured through the gastric Waterload test and the Heartbeat Counting Task, respectively. Thus, we ran two correlation analyses using a correlation package (v0.6.0; [52]) to test the relation between participants’ consistency, the Heartbeat Counting task, and the Two-step Water Load Test. ”

Data analysis, “Interoceptive accuracy and beliefs in patients” paragraph, line 234, page 11: “To test participants’ consistency in detecting the two minimum stimuli (i.e., void2 and void3) in the two groups, we performed two non-parametric t-tests.”

Results, “Relationship between the Interoceptive Tasks” paragraph, line 249, page 11:

“The correlation analyses performed considering the whole sample between participants’ consistency in detecting their minimum urinary stimuli (measured as the difference between void2 and void3) and cardiac and gastric interoceptive tasks showed that participants’ consistency in detecting their minimum urinary stimuli did not correlate with participants’ ability to detect either their gastric (R=-0.096, p=0.46) or cardiac (R=-0.03, p=0.82) signals.”

Results, “Interoceptive accuracy and beliefs in patients” paragraph, line 294, page 13: “We also performed two non-parametric t-tests to test whether void2 and void3 were consistently different in the two groups. Results showed that void2 did not significantly differ from void3 neither in the control group (W=300, p=0.171) nor in the patients' group (W=265, p=0.510).”

Consequently, we edited the Discussion to integrate the findings regarding participants’ consistency accordingly. The related discussion paragraphs at line 479, page 19, now read as follows:

“In addition to the urinary interoceptive index, this study considered a measure of consistency in perceiving the minimum interoceptive stimulus (i.e., void2 and void3). When exploring participants’ consistency, the results show that both groups did not significantly differ in the amount of urine expelled during the minimum stimuli. In contrast, control participants were more accurate in perceiving the two different urinary stimuli (i.e., void1 vs. void2 and void3) than endometriosis patients. This suggests that endometriosis pain hypervigilance may selectively affect the ability to differentiate between interoceptive stimuli of varying intensities due to an increased attentional focus on urgency or discomfort. On the contrary, the perception of lower-intensity stimuli across repeated instances remains unaffected, as these may not trigger the same heightened salience.

Moreover, urinary consistency was not correlated to performance in the cardiac and gastric interoceptive tasks, suggesting that this measure captures a distinct facet of interoception. Conversely, the urinary interoceptive index and the gastric interoceptive index are correlated. Thus, it is possible that the discrimination between two stimulus intensities, rather than consistency in detecting the same stimulus intensity, may rely on shared mechanisms across gastric and urinary domains. These findings highlight the need for future research to consider interoception as a multidimensional construct, where different components may be differentially affected in clinical conditions such as chronic pain.”

We hope that with this integration, we have now answered the reviewer’s concerns.

Reviewer #2: I have read the rebuttal and revised manuscript and am satisfied with the authors' changes.

Key concerns, including updating interoceptive terminology to align with the multidimensional framework by Suksasilp & Garfinkel (2022), clarifying the power analysis details, and providing further methodological and analytical clarifications regarding the urinary interoception task, have been adequately addressed.

The authors have also improved transparency in their reporting of the water load test, ensured inclusive language, and discussed potential limitations where appropriate.

The manuscript is engaging, and the results provide meaningful insights into interoceptive accuracy in patients with endometriosis across different bodily axes, contributing to interoception and chronic pain research.

We thank the reviewer, and we are happy that our revisions to the manuscript answered the reviewer’s concerns.

---

## [Decision Letter · Decision Letter 2]

30 Mar 2025

Impaired gastric and urinary but preserved cardiac interoception in women with endometriosis

PONE-D-24-50496R2

Dear Dr. Cantoni,

We’re pleased to inform you that your manuscript has been judged scientifically suitable for publication and will be formally accepted for publication once it meets all outstanding technical requirements.

Kind regards,

Prof. Jane Elizabeth Aspell, PhD

Academic Editor

PLOS ONE

Reviewers' comments:

Reviewer's Responses to Questions

**Comments to the Author**

1. If the authors have adequately addressed your comments raised in a previous round of review and you feel that this manuscript is now acceptable for publication, you may indicate that here to bypass the “Comments to the Author” section, enter your conflict of interest statement in the “Confidential to Editor” section, and submit your "Accept" recommendation.

Reviewer #1: All comments have been addressed

2. Is the manuscript technically sound, and do the data support the conclusions?

Reviewer #1: Yes

3. Has the statistical analysis been performed appropriately and rigorously? 

Reviewer #1: Yes

4. Have the authors made all data underlying the findings in their manuscript fully available?

Reviewer #1: Yes

5. Is the manuscript presented in an intelligible fashion and written in standard English?

Reviewer #1: Yes

6. Review Comments to the Author

Reviewer #1: My comments have been addressed. Thank you for this interesting new paper. I look forward to follow up studies addressing these limitations as I would be very excited to see the results.

7. PLOS authors have the option to publish the peer review history of their article (what does this mean? ). If published, this will include your full peer review and any attached files.

**Do you want your identity to be public for this peer review?** For information about this choice, including consent withdrawal, please see our Privacy Policy .

Reviewer #1: No

---

## [Editor Report · Acceptance letter]

PONE-D-24-50496R2

PLOS ONE

Dear Dr. Cantoni,

I'm pleased to inform you that your manuscript has been deemed suitable for publication in PLOS ONE. Congratulations! Your manuscript is now being handed over to our production team.

Kind regards,

on behalf of

Prof. Jane Elizabeth Aspell

Academic Editor

PLOS ONE